# Cervical Mesonephric Adenocarcinoma Treated with Neoadjuvant Chemotherapy: A Case Report and a Literature Review

**DOI:** 10.3390/diseases12110282

**Published:** 2024-11-07

**Authors:** Hiroaki Ishida, Megumi Manrai, Hiroki Egashira, Mizuki Nonaka, Nobuyuki Hiruta, Reiko Watanabe, Akiko Takashima

**Affiliations:** 1Department of Obstetrics and Gynecology, Toho University Medical Center Sakura, 564-1 Shimoshizu, Sakura City 285-8741, Chiba Prefecture, Japan; man-megu@sakura.med.toho-u.ac.jp (M.M.); hiroki.egashira@med.toho-u.ac.jp (H.E.); mizuki.motohashi@med.toho-u.ac.jp (M.N.); takashima-04@sakura.med.toho-u.ac.jp (A.T.); 2Department of Pathology, Toho University Medical Center Sakura, Sakura City 285-8741, Chiba Prefecture, Japan; nhr@med.toho-u.ac.jp; 3Department of Pathology, St. Marianna University of Medicine, Sakura City 285-8741, Chiba Prefecture, Japan; reikwata@gmail.com

**Keywords:** cervical mesonephric adenocarcinoma, neoadjuvant chemotherapy, adjuvant chemotherapy

## Abstract

Introduction: Cervical mesonephric ductal adenocarcinoma (MA) is an HPV-independent adenocarcinoma that occurs in middle-aged women. MA originates from remnants of the Wolffian duct that usually regress in females once the induction of sex differentiation is activated. MA is a rare disease that accounts for less than 1% of all cervical adenocarcinomas. Clinical Case: We report a case of MA in which abdominal radical hysterectomy (ARH) was performed after neoadjuvant chemotherapy (NAC). The patient was a 66-year-old woman with abnormal genital bleeding. A colposcopy examination revealed macroscopic invasive cancer. A pelvic MRI scan revealed a 53 × 26 mm tumor in the cervix, and the histological diagnosis of the cervix was endometrioid carcinoma, with the diagnosis being cervical adenocarcinoma cT1b3N0M0. One course of NAC with paclitaxel-carboplatin (PC) was administered to shrink the tumor and stop the bleeding, and ARH was performed. Postoperative histopathological diagnosis was MA. The surgical margins of the resected specimen were negative, and NAC had been effective, so the patient underwent five courses of PC therapy after surgery. There has been no recurrence 12 months after surgery. Conclusions: There is no established standard treatment, but there are reports that PC therapy is effective. It is necessary to search for effective treatments by following up and accumulating further cases.

## 1. Introduction

Mesonephric adenocarcinoma (MA) of the cervix is HPV-independent adenocarcinoma [1]. It is currently believed that MA originates from remnants of the Wolffian duct that usually regress in female once the induction of sex differentiation is activated. The average age of patients is 53 years, and it often originates from the lateral wall of the cervix [1]. MA is a rare form of cervical adenocarcinoma, accounting for less than 1% of all cervical adenocarcinomas. In a multicenter collaborative study by Pors J et al. [2], 30 cases of MA and their clinicopathological characteristics were analyzed. Follow-up data revealed that the prognosis of cervical MA was worse than that of HPV-related adenocarcinoma. No standard treatment has been established for cervical MA. We report a case of cervical MA in which radical hysterectomy was performed after neoadjuvant chemotherapy, based on a literature review.

## 2. Clinical Case

The patient was 66 years old, G2P2, and experienced menopause at 58 years old, with a medical history of diabetes.

She had been experiencing abnormal genital bleeding for the past six months, and one month ago she began experiencing daily bleeding, so she visited a local doctor. A cytological examination at a local hospital revealed cervical cytology, AGC, and endometrial cytology class IIIb, and a tumor was found in the cervix. A cervical biopsy was performed, and the patient was diagnosed with endometrioid carcinoma, and referred to our hospital. Colposcopy of the cervix at our hospital revealed a bulky tumor occupying the vagina, and the tumor was diagnosed as macroscopic invasive carcinoma (Figure 1). A transvaginal ultrasound scan revealed a cervical tumor measuring 47 × 22 mm (Figure 2), and a pelvic MRI scan revealed a cervical tumor measuring 53 × 26 mm (Figure 3A). CT scan showed no distant metastasis. Tumor markers were as follows: CA125: 13.3 U/mL; CEA: 2.4 U/mL; CA19-9: 6.5 U/L; SCC: 0.5 ng/mL; and no abnormal values were found. A histological diagnosis of the cervix at our hospital also revealed a diagnosis of endometrioid carcinoma. Based on the above, the patient was diagnosed with cervical adenocarcinoma, clinical stage IB3 (cT1b3N0M0). Surgery was scheduled for 40 days later. As the tumor was bulky and bleeding continued, one course of neoadjuvant chemotherapy with paclitaxel–carboplatin (PC) was administered with the aim of reducing the tumor and stopping the bleeding. Dose of PC therapy (paclitaxel 175 mg/m^2^, carboplatin AUC 6). After one course of PC, the cervical tumor was reduced in size (Figure 3B), and the patient was diagnosed with partial response (PR). 

Forty days after the initial consultation, an abdominal radical hysterectomy + bilateral salpingo-oophorectomy + pelvic lymphadenectomy (ARH + BSO + PLA) was performed. The operation time was 6 h and 22 min and the blood loss was 905 mL; the postoperative course was uneventful, and the patient was discharged on the ninth day after the operation. Macroscopic findings of the resected specimen showed that the resection margins between the tumor and the parametria were negative, and the resection was diagnosed as complete (Figure 4). HE staining revealed that atypical cells proliferated forming a lumen, and the lumen contained eosinophilic protein substances (Figure 5A). A tumor was found to have developed from the remnant of the mesonephric duct (Figure 5B). Immunohistochemistry showed that it was transacting T-cell-specific transcription factor GATA-3 (GATA3) (Figure 6A), clusters of differentiation 10 (CD10) (Figure 6B) and thyroid transcription factor 1 (TTF-1) positive (Figure 6C) and estrogen receptor (ER) negative, (Figure 6D). Block-positive for p16, an indicator of HPV-dependent cervical cancer, was not observed (Figure 6E). Therefore the tumor was diagnosed as HPV-independent mesonephric adenocarcinoma. The surgical margins of the resected specimen were negative, and the tumor was sufficiently separated from the lesion. As preoperative chemotherapy was effective, the patient underwent five courses of PC therapy (triweekly) after surgery. No adverse events requiring reduction or discontinuation of chemotherapy were observed. There has been no recurrence as of 12 months after surgery.

## 3. Discussion

The mesonephric duct, also known as the Wolffian duct, runs parallel to the paramesonephric duct (Mullerian duct) during early pregnancy, and in men it develops and differentiates into the male internal genital organs (epididymis and vas deferens) under the action of testosterone.

In women, unlike men, the paramesonephric ducts (Mullerian ducts) develop and differentiate to form the female internal genital organs (uterus, fallopian tubes, and upper one-third of the vagina [3]. In women, the mesonephric duct regresses, but residual tissue may be found around the ovary, within the broad ligament, uterus, and vagina [4]. Cervical mesonephric adenocarcinoma (MA) has been reported to arise mostly from the mesonephric duct located deep within the lateral wall of the cervix, but it can also arise from remnants of the mesonephric duct in the uterine body [5].

The initial symptom of cervical MA is often abnormal genital bleeding [5], and in this case, the diagnosis was made based on abnormal genital bleeding. It has been reported that the detection rate of abnormal cells in cytology is as low as 10% [5], but in this case, a solid tumor was already occupying the vagina at the time of initial diagnosis, and cervical cytology showed that AGC was diagnosed. Cervical MA has a wide variety of histological findings, and the preoperative diagnostic rate in biopsy tissue is reported to be 20% [6]. Therefore, it has been reported that it may be confused with other more common adenocarcinomas, such as serous, clear cell, or endometrial adenocarcinoma [7]. The classic histopathological pattern is the proliferation of columnar cells forming lumina containing acidophilic hyaline secretions [1]. In this case, preoperative pathological diagnosis showed that atypical cells had a glandular structure, and the case was diagnosed as cervical endometrioid carcinoma. In addition to morphological features, immunostaining is effective in diagnosing cervical MA. In tumor tissue, GATA-3, paired box 8 (PAX8), and CD10 were positive, and ER was negative. TTF-1 is rarely positive, p16 is not distributed, and HPV is usually not detected [1]. In this case, columnar cells proliferated and formed a lumen containing acidophilic hyaline-like secretions, and immunohistochemistry was positive for GATA3, PAX8, CD10, TTF-1, and negative for ER. The preoperative diagnosis was endometrioid carcinoma, but the final pathological diagnosis was MA.

There are few comprehensive reports on the prognosis of cervical MA, but Pors et al. reported the clinical characteristics in a multi-institutional study [2]. Of 30 cases of cervical MA, 25 were available for analysis; 10 cases (20%) were FIGO Stage I, and 15 cases (60%) were FIGO Stage II to IV. Lymph node metastasis was observed in 28% (4/14), and the recurrence rate was 50% (12/24), with the most common site of metastasis being the lungs 2). The 5-year progression-free survival rate was 60%, and the overall survival rate was 74%. Multivariate analysis revealed that the disease progression-free survival rate was associated with advanced stage (*p* value 0.04), but no association was observed with age, tumor size, or lymph node metastasis [2]. In a two-year follow-up study of cervical MA, HPV-dependent adenocarcinoma, and HPV-independent adenocarcinoma, the PFS of HPV-dependent adenocarcinoma was favorable (*p* value < 0.01), while no significant difference in PFS was observed between cervical MA and HPV-independent adenocarcinoma [2]. Based on these facts, cervical MA has a poorer prognosis than HPV-dependent cervical cancer, and is a pathological condition that is prone to distant metastasis, especially lung metastasis, but it is a rare disease and there is no evidence regarding treatment. There is a lack of standard treatment, and no standard treatment has been established.

We reviewed literature reporting on the treatment of cervical MA over the past 10 years from 2014 to 2024 (Table 1) [4,8,9,10,11,12,13,14,15,16]. Neoadjuvant chemotherapy (NAC) was administered to three patients, including this case, and all were partial responses, and radical hysterectomy was subsequently performed. The initial treatment for all 12 patients was total hysterectomy, and as postoperative adjuvant therapy, 4 patients underwent CCRT and 1 patient received radiation therapy alone. Chemotherapy was administered in three patients, and no post-treatment was administered in four patients. The effect of NAC in improving prognosis in stage I-II cervical cancer is unclear [17]. However, NAC is being considered as an option for patients undergoing elective treatment for cervical cancer during the COVID-19 pandemic, according to the International Gynecologic Cancer Society (IGCS) and the European Society of Gynecological Oncology (ESGO) [18]. In this case, the preoperative pathological diagnosis was endometrioid carcinoma, which is HPV-independent, and radiation therapy may have been ineffective. In addition, the bulky tumor occupied the vagina and there was a lot of bleeding, so early treatment was necessary. Therefore, PC therapy was administered as NAC, and since the tumors were partial responded and bleeding decreased, a radical hysterectomy was performed after one course of chemotherapy was completed. Although the number of cases of NAC is limited, all three reported cases were successful, and therefore we believe that NAC should also be considered for cervical MA. Concerning postoperative adjuvant therapy for cervical cancer, CCRT is recommended for cases at risk of recurrence [19]. On the other hand, in a clinical trial investigating the effectiveness of radiation alone, CCRT, and chemotherapy as postoperative therapy for stage IB-II cervical adenocarcinoma, it was reported that there was no difference in survival rate among the three groups, and chemotherapy may also be effective as a postoperative adjuvant therapy [20]. Although no clear standard chemotherapy has been established for cervical MA, there have been reported cases in which PC therapy was effective in recurrent cases [21]. Considering these circumstances, in this case, because NAC had been effective and the tumor resection margins were adequately secured, chemotherapy with PC therapy was selected as postoperative adjuvant therapy.

## 4. Conclusions

We report a case of cervical mesonephric adenocarcinoma in which preoperative chemotherapy was effective. Cervical MA is very rare, and preoperative diagnosis is difficult. Because it is a rare disease, there is currently no established standard treatment, but it has been suggested that PC therapy may be effective. This disease is characterized by a high incidence of distant metastasis to the lungs and other organs, and it is necessary to conduct long-term follow-up and accumulate more cases to find effective treatments.

## Figures and Tables

**Figure 1 diseases-12-00282-f001:**
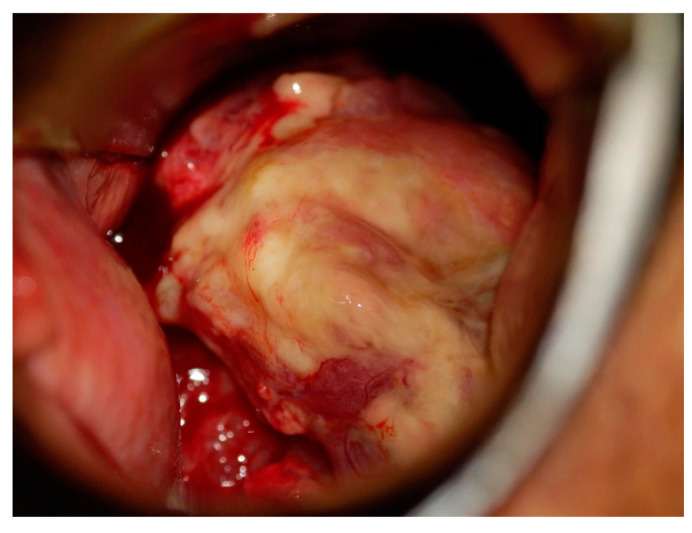
Colposcopy of the cervix. A bulky tumor occupying the vagina, and the tumor was diagnosed as macroscopic invasive carcinoma.

**Figure 2 diseases-12-00282-f002:**
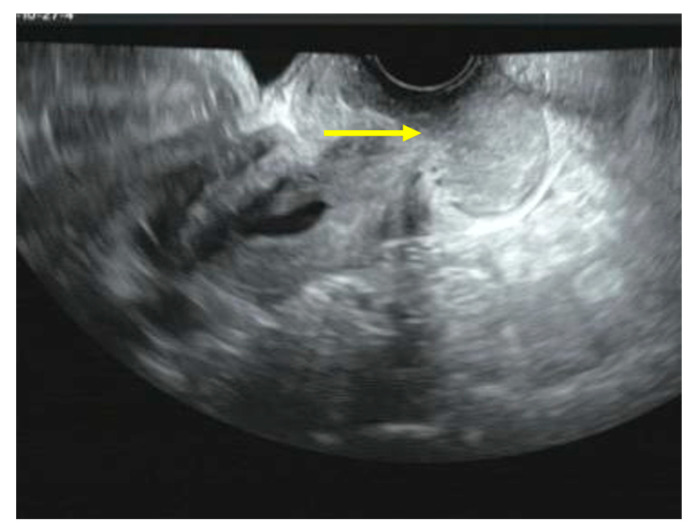
Transvaginal ultrasound scan. A cervical tumor measuring 47 × 22 mm.

**Figure 3 diseases-12-00282-f003:**
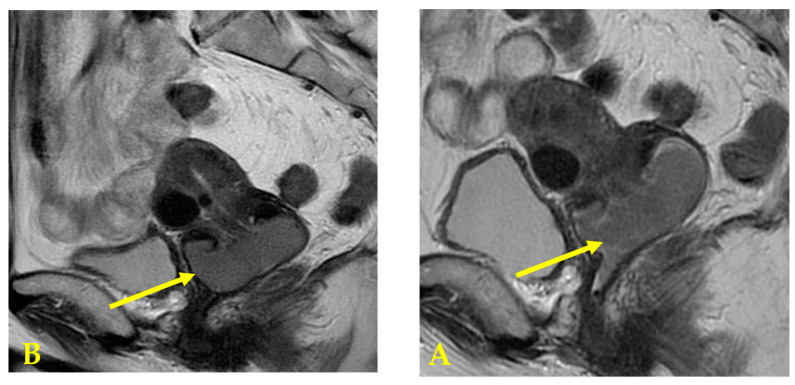
(**A**) Pelvic MRI T2 weighted image: A cervical tumor measuring 53 × 26 mm. Before chemotherapy. (**B**) Pelvic MRI T2 weighted image: After one course of neoadjuvant chemotherapy with paclitaxel–carboplatin (PC). A cervical tumor measuring 43 × 23 mm. The tumor shrank and PR was diagnosed.

**Figure 4 diseases-12-00282-f004:**
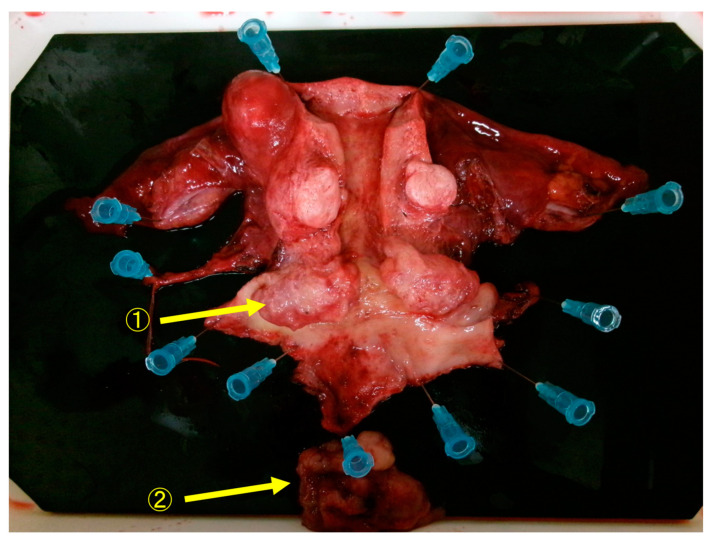
Macroscopic findings of the resected specimen showed that the resection margins between the tumor and the parametria were negative, and the resection was diagnosed as complete. ① Cervical tumor. ② Tumor that has fallen off from the cervical tumor.

**Figure 5 diseases-12-00282-f005:**
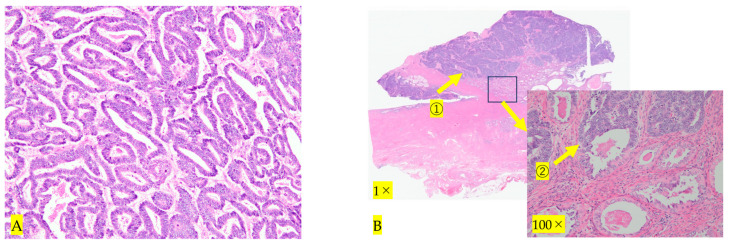
(**A**) **HE staining at 100×.** Atypical cells proliferate and form a lumen that contains eosinophilic protein material. (**B**) **HE staining at 1× 100×.** Atypical cells proliferate (①→). Carcinoma originating from the mesonephric duct (②→).

**Figure 6 diseases-12-00282-f006:**
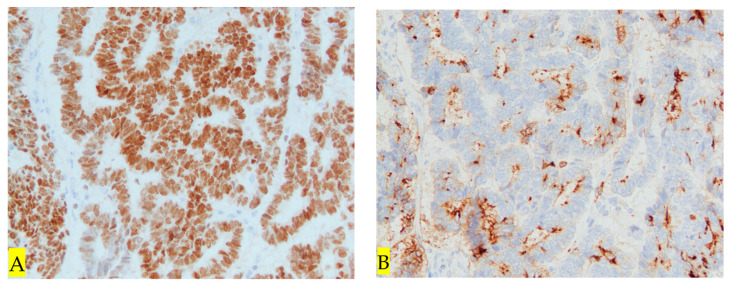
(**A**) **Immunostaining: GATA at 200×.** GATA positivity in tumor tissue. (**B**) **Immunostaining: CD10 at 100×.** CD10 positivity found in the stroma. (**C**) **Immunostaining: TTF-1 at 200×.** TTF-1-positive in some tumor tissues. (**D**) **Immunostaining: ER at 1×.** ER-negative in tumor tissue (→). (**E**) **Immunostaining: p16 at 1×.** Block-positive for P16 is negative in tumor tissue (→).

**Table 1 diseases-12-00282-t001:** Literature reporting on the treatment of cervical MA over the past 10 years from 2014 to 2024.

Case	Age	Stage	NAC	Operation	Adjuvant Therapy	PFS Months	Out Come
Nili F [8]	46	IB3	—	ATH tubectomy	CCRT with CCDP	9 m	Peritoneal dissemination
Reis-de-Carvalho C [9]	60	IB1	—	ARH BSO PLND	CCRT with CDDP	60 m	No recurrence
Marani C [10]	58	IB2	—	ATH BSO	—	72 m	Vaginal recurrence
Jiang LL [11]	48	IVB	—	ARH BSO PLND Omentectomy appendectomy	Chemotherapy	32 m	Lung recurrence
Montalvo N [12]	48	IB1	—	ATH	—	36 m	Lung recurrence
Papoutsis D [13]	67	IIB	—	ARH BSO PLND	CCRT and brachytherapy	12 m	No recurrence
Abdul-Ghafar J [14]	48	IB2	—	Vaginal hysterectomy	CCRT	24 m	No recurrence
Xie C Case1 [4]	50	IB1	—	ARH BSO	—	64 m	No recurrence
Xie C Case2 [4]	49	IB1	—	ARH BSO LND	—	70 m	No recurrence
Ditto A [15]	51	IIB	CDP	ARH BSO PLND	RT	6 m	No recurrence
Kuratsune K [16]	30	IIA2	PC 2kur	ARH BSO PLND	PC 4kur	13 m	No recurrence
This Case	66	IB3	PC 1kur	ARH BSO PLND	PC 5kur	12 m	No recurrence

ATH: abdominal total hysterectomy; CCRT: concurrent chemoradiotherapy; CDDP: cisplatin; ARH: abdominal radical hysterectomy; BSO: bilateral salpingo-oophorectomy; PLND: pelvic lymphadenectomy; LND: lymph node dissection; CDP: cisplatin doxorubicin paclitaxel; PC: paclitaxel carboplatin.

## Data Availability

Datasets generated and/or analyzed during the current study are available from the corresponding author upon reasonable request.

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
