# Peer review of "Cervical Mesonephric Adenocarcinoma Treated with Neoadjuvant Chemotherapy: A Case Report and a Literature Review"

_diseases, 2024, doi:10.3390/diseases12110282_

Round 1

Reviewer 1 Report

Comments and Suggestions for Authors

Interesting case. However, please revise the manuscript according to CARE case report guidelines (CARE Case Report Guidelines (care-statement.org)).

Author Response

Comment1:However, please revise the manuscript according to CARE case report guidelines (CARE Case Report Guidelines (care-statement.org)).

Response:  
CARE  9b Administration of therapeutic intervention (such as dosage, strength, duration) .

Since the above information was not included, the following text was added to line 58: "Dose of PC therapy (paclitaxel 175 mg/m2, carboplatin AUC 6)"

CARE 10d Adverse and unanticipated events . . . . . . . . . . . . . . . . . . . . . . . . . . . . . . . . . . . . . . . . . . . . . . . . . .

Since the above information was not included, the following text was added to line 76-78: "the patient underwent five courses of PC therapy (triweekly) after surgery. No adverse events requiring reduction or discontinuation of chemotherapy were observed. "

The corrected areas are highlighted in yellow.

Reviewer 2 Report

Comments and Suggestions for Authors

Thu authors present a case report, and it was about cervical mesonephric adenocarcinoma treated with neoadjuvant chemotherapy.

This tpye of cervical carcinoma is rare, and the author review the previous paper and summarize the therapy strategy of them.  From the MRI imaging, the tumor shrinked after chemotherapy. And they make the surgery.

I think that this case is attractive, and it might be interesting to the audence.

Comments on the Quality of English Language

I think that  authors should optimize Englishi writing.

Author Response

Comments : I think that  authors should optimize Englishi writing.

Response :  Editage helped with English proofreading.

The corrected areas are highlighted in yellow.

Reviewer 3 Report

Comments and Suggestions for Authors

The paper proposed by Hiroaki Ishida et al., and entitled “Cervical mesonephric adenocarcinoma treated with neoadjuvant chemotherapy. A Case report and a literature review” reports a case of cervical adenocarcinoma diagnosed as a Mesonephric adenocarcinoma (MA). Since this is a rare histological form of cervical carcinoma, the authors propose their observation with detailed therapeutic procedure and favorable outcome for publication.

Major remark: Overall, the paper is well documented. However, it is difficult to make the diagnosis on the pictures proposed. Even if figure 5B shows some tubes that might correspond to remnants of the mesonephric tube, this is not sufficient to support the diagnosis of MA (remnants could be associated with endometrioid carcinoma as well). In other terms, the evidence that the present tumor is HPV negative is lacking in the observation. Such a negativity would be a weak additive diagnostic criterion, but in the other hand, HPV positivity would necessarily conduct to discard the diagnosis of MA.

In conclusion, the result of HPV genotyping is a necessary step to include in this report. At least use of standard PCR is recommended. The negativity of p16 staining is not sufficient to support HPV negative result, but could also be presented.

Minor remark: Do you think that the statement that MA “originates from remnants of the wolffian duct in the male reproductive system during fetal development” is appropriate? It might be more appropriate to better explain that wolffian duct usually regress in female once the induction of sex differentiation is activated.

Comments on the Quality of English Language

No specific remark

Author Response

Comments 3 Major remark: Overall, the paper is well documented. However, it is difficult to make the diagnosis on the pictures proposed. Even if figure 5B shows some tubes that might correspond to remnants of the mesonephric tube, this is not sufficient to support the diagnosis of MA (remnants could be associated with endometrioid carcinoma as well). In other terms, the evidence that the present tumor is HPV negative is lacking in the observation. Such a negativity would be a weak additive diagnostic criterion, but in the other hand, HPV positivity would necessarily conduct to discard the diagnosis of MA.

In conclusion, the result of HPV genotyping is a necessary step to include in this report. At least use of standard PCR is recommended. The negativity of p16 staining is not sufficient to support HPV negative result, but could also be presented.

Response  Thank you for pointing that out.

Figure 5B has been changed to a slide showing cancer developing in the mesonephric duct. The text on lines 71-72 has been revised as follows: A tumor was found to have developed from the remnant of the mesonephric duct (Figure5B).

HPV genotyping was not performed.
Immunostaining for p16 was performed, but no p16 block positivity was observed.        Figure 6E (Immunostaining p16 )has been added.                                      The following text has been added to lines 76 to 78:                                        Block-positive for p16, an indicator of HPV-dependent cervical cancer, was not observed (Figure 5E). Therefore the tumor was diagnosed as HPV-independent mesonephric adenocarcinoma.

Comments 3 Minor remark: Do you think that the statement that MA “originates from remnants of the wolffian duct in the male reproductive system during fetal development” is appropriate? It might be more appropriate to better explain that wolffian duct usually regress in female once the induction of sex differentiation is activated.

Response  Thank you for pointing that out.

The sentences on lines 13 and 34 have been corrected as pointed out.

“that usually regress in female once the induction of sex differentiation is activated ”

The corrected parts are highlighted in yellow

Round 2

Reviewer 1 Report

Comments and Suggestions for Authors

Thanks for revising the manuscript. I have no more questions.

Reviewer 3 Report

Comments and Suggestions for Authors

Thank you for this paper.

p16 negativity is a weak argument for retaining the diagnosis of HPV negative tumor.

Comments on the Quality of English Language

No specific comment